# Exploration of Postharvest Conditions for *Codonopsis pilosula* Nannf. var. *modesta* (Nannf.) L. T. Shen Roots Based on Sensory Quality, Active Components, Antioxidant Capacity and Physiological Changes at Different Storage Temperatures

**DOI:** 10.3390/foods12244418

**Published:** 2023-12-09

**Authors:** Longxia Wen, Yanping Wang, Pingping Song, Zixia Wang, Zhuoshi Tang, Yina Guo, Huaqiao Yu, Fangdi Hu

**Affiliations:** School of Pharmacy, Lanzhou University, Lanzhou 730000, China; wenlx21@lzu.edu.cn (L.W.); ypwang13@lzu.edu.cn (Y.W.); songpp2020@lzu.edu.cn (P.S.); wangzx20@lzu.edu.cn (Z.W.); tangzhsh21@lzu.edu.cn (Z.T.); guoyn21@lzu.edu.cn (Y.G.); yuhq21@lzu.edu.cn (H.Y.)

**Keywords:** *Codonopsis pilosula* Nannf. var. *modesta* (Nannf.) L. T. Shen, temperature, sensory quality, active components, antioxidant capacity, postharvest physiology

## Abstract

The promotion of industrial-mode production of *Codonopsis pilosula* Nannf. var. *modesta* (Nannf.) L. T. Shen (*C. pilosula*) has expanded the demand for the postharvest storage of fresh roots. Further research is needed to establish comprehensive methods to evaluate the impact of storage conditions. This study simulated the storage process of roots at near-freezing temperature [NFT (−1 °C)] and traditional low temperatures (−6 °C, 4 °C and 9 °C) for 40 days. At different storage stages, correlation analysis was conducted using quantitative data on 20 parameters, including sensory quality, active components, antioxidant capacity and physiological changes. Appearance and principal component analysis could distinguish between fresh and stored samples, while NFT samples on the 40th day of storage were similar to fresh ones. Correlation analysis indicated that NFT storage could maintain the sensory quality by increasing the antioxidant enzyme activity and active components, reducing the accumulation of reactive oxygen species and malondialdehyde and reducing the activity of browning-related enzymes and cell-wall-degrading enzymes. These findings highlight the importance of the overall quality evaluation of fresh roots and emphasize the potential to improve fresh root and dried medicinal material quality by regulating storage conditions such as temperature.

## 1. Introduction

*Codonopsis pilosula* Nannf. var. *modesta* (Nannf.) L. T. Shen (*C. pilosula*) is a famous “medicine food homology” of the Campanulaceae family. The root is the main medicinal and edible part, which has various active components such as sugars, flavonoids, terpenoids and saponins [1]. Dried *C. pilosula* roots have various beneficial effects such as improving the body’s immunity, regulating gastrointestinal motility and reducing blood pressure [2]. They are widely demanded as food materials, dietary supplements and traditional Chinese medicine in China, Japan, Singapore and other Asian countries [3,4]. Dried roots and their decoction pieces are currently the main forms for sale. However, traditional processing methods usually require more than 2 months to fully dry them before transporting them to retailers or factories. During the drying process, in environments with low temperatures and high humidity, the roots are susceptible to infection by endogenous and exogenous pathogens after harvesting, which not only seriously affects the quality and reduces their medicinal value [5], but also leads to the accumulation of fungal toxins, ultimately threatening human health [6]. With the gradual formation of standardized, industrialized and large-scale production and processing models of *C. pilosula*, a large number of fresh roots are collected after centralized harvesting in the autumn. Thus, postharvest storage of fresh roots becomes more important.

Low-temperature storage is the most common and effective way to extend the shelf life and maintain quality of crops after harvest [7,8]. However, low temperatures may easily lead to crop frostbite. Near-freezing temperature (NFT) storage is a non-freezing technique that maintains the freshness of postharvest crops at the temperature, which is close to the physiological freezing points of their individual materials [9]. Liu B et al. [10] found that NFT storage can inhibit the production rate of superoxide anions (O_2_^−^∙), hydrogen peroxide (H_2_O_2_) content and malondialdehyde (MDA) content, and regulate the activity of superoxide dismutase (SOD), catalase (CAT) and peroxidase (POD). NFT storage can also improve shelf life by inhibiting the respiratory metabolism and reducing the activity of cell-wall-degrading enzymes such as pectin methylesterase (PME), polygalacturonase (PG) and cellulase (Cx) [7]. However, to our knowledge, no systematic studies have been reported on the effects of different temperatures, including NFT, on the storage quality of postharvest *C. pilosula* roots. The exploration of systematic evaluation methods for suitable storage conditions of fresh *C. pilosula* roots is still very lacking.

At present, a combination of chemometrics and quantitative determination of active components is the most accurate and feasible method for comprehensively evaluating and distinguishing the quality of traditional Chinese medicine [11,12]. Principal component analysis (PCA) and orthogonal partial least squares discriminant analysis (OPLS-DA) are the most common multivariate techniques [13]. Lobetyolin, syringin and atractylenolide III are the main active components in *C. pilosula* [14,15,16]. In addition, flavonoids and phenols are the main antioxidant substances for medicinal plants to exert antioxidant activity and resist postharvest stress. DPPH radicals are a relatively stable type of free radical, and their scavenging ability is a commonly used indicator for the in vitro evaluation of plant antioxidant activity [17]. The above overall quality control methods and the selection of relevant quality parameters for *C. pilosula* medicinal materials have important guiding significance for the quality control of harvested roots in this study.

In this experiment, the storage process of fresh *C. pilosula* roots under NFT and traditional low-temperature storage conditions (−6 °C, 4 °C and 9 °C) for 40 days was simulated. The purpose of this study is to: (i) monitor the sensory quality, effective components contents, antioxidant capacity and physiological quality changes in *C. pilosula* roots under different storage temperatures; (ii) establish and evaluate the comprehensive quality characteristics of roots at different storage stages using PCA and OPLS-DA; (iii) evaluate the interaction between different quality parameters using correlation analysis of quantitative data.

## 2. Materials and Methods

### 2.1. Plant Materials and Experimental Design

Four-year-old *Codonopsis pilosula* Nannf. var. *modesta* (Nannf.) L. T. Shen roots were collected from Haxigou Village, Zhongzhai Town, Wen County, Longnan City, Gansu Province (104° 28′ E, 33°16′ N), in October. After collection, the samples in soil were immediately transported back to the laboratory (1–2 d of transportation). Samples with an intact appearance, no insect infestation or mold, no damage and a uniform size were selected. The roots were washed with water. After air-drying, they were placed in a PA/PE microporous preservation bag (20 cm × 45 cm, thickness 0.1 mm, with 8 pairs of holes with a pore size of 0.1 mm each at a distance of 5 cm from both sides of the bag). Each bag contained 6 root samples (approximately 300 g) and was pre-cooled at 4 °C for 24 h. The pre-treated samples were randomly divided into four treatment groups and stored at different temperatures (−6 °C, NFT, 4 °C and 9 °C) with a humidity of 70 ± 5% for 40 d. During the storage process, samples were observed every 20 d, and 18 roots (approximately 900 g) per treatment group were sampled per time, with 6 roots serving as biological replicates. After shredding and mixing, one portion was frozen in liquid nitrogen and stored at −80 °C for enzyme activity measurement, while the other portion was freeze-dried in vacuum, crushed and stored at −20 °C for chemical analysis.

### 2.2. Freezing Point Temperature of C. pilosula Roots

The freezing method was used to determine the freezing point temperature [18]. Fresh roots were ground and filtered with a double layer of gauze. The filtrate was placed in a refrigerator at −18 °C. When it was stirred continuously, the temperature was recorded every 30 s. After the filtrate was completely frozen, a temperature change curve was established to determine the freezing point temperature.

### 2.3. Storage Effect and Quality

#### 2.3.1. Sensory Analysis

The treated roots were evaluated by 30 trained participants (50% female and 50% male) aged 22–50 years. According to Salas-Sanjuán et al. [19], before the experiment, the panelists were trained by evaluating highly favorable sensory attributes of *C. pilosula* roots, such as appearance, odor, texture and taste.

According to the sensory characteristics of the *C. pilosula* roots, we used a 1–25 scale where 25 represents high quality and 1 represents low quality for appearance, odor, texture and taste. Each panelist evaluated 6 representative samples from each treatment and sampling time. The evaluation score comprised the sum of the scores of the four sensory attributes.

#### 2.3.2. Weight Loss and Respiratory Intensity

The weight loss was calculated using the weighing method. The initial weight of each group was recorded as $m_{1}$/g, and the measured mass was recorded as $m_{2}$/g every 20 d. The weight loss (%) was calculated based on the equation below (1).
(1)$$\mathrm{weight}\;\mathrm{loss}\;(\%)=\frac{m_{1}-m_{2}}{m_{1}}\times 100$$

The respiratory intensity was measured using a respiration meter (3051H, Topp Yunnong Technology Co., Ltd., Hangzhou, China). According to Lin et al. [20], the results were calculated based on the mass of CO_2_ released per kilogram of sample per hour at 20 °C. Three representative samples were randomly selected from each group, weighed and placed in a 2.0 L sealed container at 20 °C for 1 h. The average respiratory intensity was calculated using the instrument’s own software.

#### 2.3.3. O_2_^−^∙ Production Rate, H_2_O_2_ Content and MDA Content

Wang, H.F.’s method [21] was modified to measure the production rate of O_2_^−^∙. Thus, 1.5 mL of phosphate buffer (50 mmol·L^−1^, pH 7.8) was added to 0.3 g of ground root sample, extracted at 4 °C for 30 min and then centrifuged for 10 min (8000× *g*, 4 °C). After centrifugation, 1.0 mL of phosphoric acid buffer (50 mmol·L^−1^, pH 7.8) and 1.0 mL of hydroxylamine hydrochloride (10 mmol/L) were added to 0.8 mL of the supernatant, and the mixture was placed in a 25 °C water bath for 20 min. Then, 1.0 mL of p-aminobenzene sulfonic acid (17 mmol/L) and 1.0 mL of α-naphthylamine (7 mmol/L) were added to the above mixture to react in a constant temperature water bath at 25 °C for 30 min, and then centrifuged again. The absorbance of the supernatant was measured at 530 nm using an ultraviolet spectrophotometer (UV-1700, Shimadzu Co., Kyoto, Japan). The rate of O_2_^−^∙ production was calculated according to the equation below (2), expressed as the amount of O_2_^−^∙ substance produced per kilogram of fresh weight sample per minute (µmol·min^−1^·kg^−1^). Potassium nitrite was used as a standard solution (100 µmol/L). Taking the amount equivalent to the O_2_^−^∙ substance (nmol) as the abscissa (X) and the A530 value as the ordinate (Y), the standard curve Y = 5.1232X − 0.0006 (*R*^2^ = 0.9998) was obtained, with a linear range of 0~0.12 nmol.
(2)$${O_{2}}^{-}\cdot \;\mathrm{production}\;\mathrm{rate}\;(µ\mathrm{mol}\cdot \min^{-1}\cdot {\mathrm{kg}}^{-1})=\frac{n\times V_{1}\times 1000}{V_{S}\times T\times m}$$

n represents the amount of O_2_^−^∙ substance in the solution converted from the standard curve (µmol), $V_{1}$ represents the volume of the sample extraction solution (mL), $V_{S}$ represents the volume of the sample extraction solution taken during measurement (mL), T represents the reaction time between the sample and hydroxylamine hydrochloride (min) and m represents the fresh weight of the sample (kg).

The determination of H_2_O_2_ content should refer to Cao, J.K.’s method [22] and be modified. Thus, 0.5 g of root sample was homogenized with 1.5 mL of acetone, and then centrifuged at 4 °C for 10 min (8000× *g*). Then, 0.1 mL of titanium sulfate (5%) and 0.1 mL of concentrated ammonia water were added to 1 mL of the supernatant, which was mixed and centrifuged again for 10 min (8000× *g*). The precipitate was washed three times with acetone and then dissolved in 5 mL of sulfuric acid (2 mol/L). After dissolution, the absorbance of the supernatant was measured at 415 nm using an ultraviolet spectrophotometer (UV-1700, Shimadzu Co., Japan). The H_2_O_2_ content was calculated according to the equation below (3), expressed as the amount per kilogram of fresh tissue (mmol·kg^−1^). H_2_O_2_ was used as a standard solution (100 µmol/L). The equivalent to the amount of H_2_O_2_ (µmol) represents the abscissa (X) and A415 represents the ordinate (Y), resulting in a standard curve Y = 0.0775X + 0.0003 (*R*^2^ = 0.9991) with a linear range of 0–10 µmol.
(3)$$H_{2}O_{2}\;\mathrm{content}\;(\mathrm{mmol}\cdot {\mathrm{kg}}^{-1})=\frac{C\times V_{1}}{m\times V_{t}}$$

C represents the concentration of H_2_O_2_ in the sample obtained from the standard curve conversion (mmol); $V_{1}$ represents the volume of sample extraction solution used in the determination (mL); m represents the fresh weight of the sample (kg) and $V_{t}$ represents the total volume of the sample extraction solution (mL).

The MDA content was determined according to the instructions of the reagent kit (Keming Biotechnology Co., Ltd., Suzhou, China). For MDA content measurement, the root tissue (0.1 g) was homogenized at 4 °C with 1 mL of cold extraction buffer and centrifuged (4 °C, 8000× *g*, 10 min). Then, 0.1 mL of the supernatant was mixed with 0.3 mL of reagent 1. The mixture was kept in a 95 °C water bath for 30 min, cooled in an ice bath and then centrifuged (25 °C, 8000× *g*, 10 min). The absorbance at 532 nm and 600 nm was measured using an ultraviolet spectrophotometer (UV-1700, Shimadzu Co., Kyoto, Japan).

#### 2.3.4. SOD, CAT, POD and PPO Activity

Thr SOD, CAT, PPO and POD activity were determined according to the instructions of the reagent kit (Keming Biotechnology Co., Ltd., Suzhou, China). The root tissue (0.1 g) was homogenized at 4 °C with 1 mL of special extraction buffer in each reagent kit and centrifuged (4 °C, 8000× *g*, 10 min), respectively. The supernatant was collected as a crude extraction.

For SOD activity measurement, the reaction mixture consisted of reagent 1 (45 μL), 18 μL of supernatant (18 μL of distilled water for the control), reagent 2 (2 μL), reagent 3 (35 μL) and reagent 4 (300 μL). The mixture stood at room temperature for 30 min. The absorbance of the sample at 560 nm was measured using a microplate reader (Tecan Spark, TecanTrading Co., Shanghai, China). The SOD activity was defined as follows: a unit of SOD activity (U) corresponded to the content of SOD when the inhibition rate of SOD reduced to 50%.

For CAT activity measurement, 10 µL of crude CAT extract was added to 190 µL of working fluid, and the changes in the absorbance of the mixture at 240 nm were measured using a spectrophotometer (UV-1700, Shimadzu Co., Kyoto, Japan). The catalytic degradation of 1 μmol of H_2_O_2_ per kilogram of tissue per minute is defined as an enzyme activity unit (U).

For POD activity measurement, 10 μL of the supernatant was mixed with 190 μL working fluid. The change in the mixture’s absorbance at 470 nm was recorded once every 1 min using a spectrophotometer (UV-1700, Shimadzu Co., Kyoto, Japan). One unit (U) was defined spectrophotometrically as an increase of 0.01 in absorbance per minute per gram of tissue in the reaction system per milliliter.

For the PPO activity analysis, 50 μL of the supernatant (50 μL of boiled samples for the control) was mixed with 200 μL of reagent 1 and 50 μL of reagent 2. The mixture was kept in a 25 °C water bath for 10 min to react and then kept in a 95 °C water bath for 5 min to stop the reaction. After cooling to room temperature, the mixture was centrifuged (25 °C, 8000× *g*, 10 min). The absorbance at 525 nm was measured using a microplate reader (Tecan Spark, TecanTrading Co., Shanghai, China). One unit (U) was defined spectrophotometrically as an increase of 0.01 in absorbance per minute per gram of tissue in the reaction system per milliliter.

#### 2.3.5. PME, PG, Cx and β-Glu Activity

We referred to the method of Ge, Y. [23] for determination. Using the amount of glucose (mg) as the x-axis (X) and the A540 value as the y-axis (Y), the standard curve Y = 0.6335X − 0.0068 (*R*^2^ = 0.9992) was obtained, with a linear range of 0–1.6 mg. The PG activity was expressed as the amount of enzyme required to release 1 mg of galacturonic acid per 1 g of fresh weight per 1 h, which was one enzyme activity unit (U·g^−1^). The PME activity was also expressed as the amount of enzyme required to release 1 mg of galacturonic acid per 1 g of fresh weight per 1 h, which was one enzyme activity unit (U·g^−1^). Similarly, the Cx activity was expressed as the amount of enzyme required to release 1 mg of reducing sugar per 1 g of fresh weight per 1 h, which was one enzyme activity unit (U·g^−1^), while the β-Glu activity was also expressed as an enzyme activity unit (U·g^−1^) corresponding to releasing 1 mg of reducing sugar per 1 g of fresh weight per 1 h.

#### 2.3.6. Lobetyolin Content, Syringin Content and Atractylenolide III Content

The *C. pilosula* samples were crushed using a pulverizer and passed through a No. 4 sieve (250 µm). Approximately 4 g of each powdered sample was extracted with 50 mL of methanol under an ultrasound for 45 min. After centrifugation at 4000× *g* for 10 min, the supernatant was concentrated and diluted to 10 mL with methanol. All of the solution was filtered through a 0.45 µm filter membrane prior to high-performance liquid chromatography (HPLC) analysis. The HPLC analysis was performed on an Agilent 1260 HPLC system (Agilent Technologies, Santa Clara, CA, USA) equipped with a diode array detector (DAD). The chromatogram column was a C18 column (4.6 mm × 250 mm, 5 µm; Eclipse XDB; Agilent, Santa Clara, CA, USA) with the column temperature set at 30 °C.

The elution conditions of lobetyolin and atractylenolide III were described according to Sisi et al. [24]. With slightly modification, the specific steps were as follows. The flow rate was 1.0 mL/min and the injection volume was 10 μL. When determining the content of lobetyolin, the mobile phase consisted of acetonitrile (A)–water (B) = 26:74 with isocratic elution. The wavelength was 267 nm. When determining the content of atractylenolide III, the mobile phase consisted of acetonitrile (A)–water (B) = 71:29 with isocratic elution. The wavelength was 220 nm. Their results were expressed as the content per kilogram of dry weight (g·kg^−1^).

The elution condition of syringin was described according to Wang et al. [25]. The mobile phase consisted of acetonitrile (A)–water (B) with gradient elution. The elution procedure was 0~10 min, 10 % (A); 10~20 min, 15 % (A); 20~30 min, 10 % (A). The wavelength was 270 nm. The flow rate was 1.0 mL/min, and the injection volume was 10 μL. The result was expressed as the content per kilogram of dry weight (mg·kg^−1^).

#### 2.3.7. Total Flavonoid Content, Total Phenol Content and DPPH Radical Scavenging Rate

The powder sample (2 g) was extracted twice with 25 mL of ethanol (70%) for 30 min each time. The filtrate was combined and diluted to 100 mL, which could be used to determine the total flavonoid content, total phenol content and DPPH radical scavenging rate.

The total flavonoid content and the total phenol content were determined using the method of Wang et al. [26], with the results expressed as the content per kilogram of dry weight (g·kg^−1^).

The DPPH radical scavenging rate was measured according to Fei et al. [27], and the specific steps were as follows: first, 1 mL of the sample solution and 1 mL of the DPPH solution (80 μg/mL) were shaken in a tube and reacted for 30 min in dark. The absorbance was evaluated at 517 nm using an ultraviolet spectrophotometer (UV-1700, Shimadzu Co., Kyoto, Japan). The DPPH radical scavenging rate was calculated according to the equation below (4).
(4)$$\mathrm{DPPH}\;\mathrm{radical}\;\mathrm{scavenging}\;\mathrm{rate}\;(\%)=\left(1-\frac{A_{1}}{A_{0}}\right)\times 100$$

$A_{1}$ and $A_{0}$ are the absorbance of the DPPH radicals without and with the sample, respectively.

### 2.4. Statistical Analysis

All the samples were subjected to three repeated analyses, and the results were expressed as mean ± standard deviation. IBM SPSS 26.0 (SPSS Inc., Chicago, IL, USA) was used for analysis of variance (ANOVA), and the differences between means were evaluated using Dunnett’s test. There was a significant difference at *p* < 0.05. PCA was conducted using the Omic Studio software (https://www.omicstudio.cn (accessed on 13 July 2023)). OPLS-DA was conducted using SIMCA 14.1(Umetrics, Umea, Sweden). Origin 2021 (Origin Lab Corporation, Northampton, MA, USA) was used for the clustering heatmap and Pearson’s correlation analysis.

## 3. Results

### 3.1. The Freezing Point Curve of Fresh C. pilosula Roots

The different content of sugars, minerals, organic acids and other substances in different crop tissues result in different freezing points. Because NFT is very close to the freezing point of fresh products, the large range of temperature fluctuations during storage may cause frostbite. Determination of the freezing point temperature is an important basis for the storage of fresh *C. pilosula* roots. After placing the filtrate of the root tissue homogenate into a −18 °C refrigerator, with continuous stirring, the internal temperature of the filtrate quickly dropped to the super-cooling point (−2.1 °C). And at this point, freezing phenomena began to occur (Appendix A). At the same time, the root tissue released heat to increase the temperature, and the temperature remained unchanged for a period of time. This temperature was the physiological freezing point of fresh *C. pilosula* roots (−1.2 °C). Considering the individual differences and fluctuations in refrigeration temperature (±0.3 °C), this experiment used −1 °C as the NFT storage temperature in order to prevent frostbite.

### 3.2. Storage Quality of C. pilosula Roots

#### 3.2.1. Changes in Appearance, Sensory Scores and Weight Loss

*C. pilosula* is a popular food and traditional Chinese medicine. Its sensory qualities, including appearance, odor, hardness and taste, determine its shelf life and product value. As shown in Figure 1A, the fresh roots appeared light yellow brown with a crisp and tender texture. But at −6 °C, the roots appeared light brown on the 20th day of storage, with a slightly wrinkled surface. On the 40th day, they turned dark brown, with a soft texture and concave surface. On the 20th day of storage at 9 °C, the roots appeared light brown in appearance, but the surface began to mold. On the 40th day, it turned gray brown and began to rot. Thus, 4 °C and NFT can effectively delay the decline in sensory quality, presenting better sensory effects, such as brighter colors, the original particular aromas, hard and elastic texture and a crispy and tender taste. Therefore, the sensory scores of the NFT roots during the entire storage period were significantly higher than those of other groups (*p* < 0.05) (Figure 1B).

The weight loss directly affects the quality and aging of crops during storage. The weight loss of roots at 9 °C increased faster than at −6 °C, NFT and 4 °C, reaching 9.31% on the 40th day of storage (Figure 1C).

#### 3.2.2. Changes in Respiratory Intensity, O_2_^−^∙ Production Rate, H_2_O_2_ Content and MDA Content

Respiratory function is closely related to the decomposition and metabolism of nutrients, which is an important cause of tissue damage and quality deterioration during postharvest storage of crops [7]. Compared with NFT and 4 °C roots, the respiratory intensity of the 9 °C roots remained at a higher level during storage, while the −6 °C roots remained at a lower level (Figure 2A).

Reactive oxygen species (ROS), including O_2_^−^∙ and H_2_O_2_, cause oxidative damage to membrane lipids and are widely believed to be the main cause of plant cell aging and apoptosis [28]. Vigorous respiration can cause the excessive generation of ROS. As shown in Figure 2B,C, the rate of O_2_^−^∙ production and H_2_O_2_ content had the same trend of change throughout the storage period, both showing a linear increase. The O_2_^−^∙ production rate and H_2_O_2_ content of roots at 9 °C were significantly higher than those at NFT and −6 °C during storage (*p* < 0.05). MDA is the final product of lipid oxidation, reflecting the degree of membrane damage caused by low-temperature or oxidative damage. On the 20th day of storage, the MDA content of the −6 °C roots was significantly higher than that of the NFT, 4 °C and 9 °C groups (*p* < 0.05).

On the 40th day of storage, the MDA content of the roots at 9 °C and −6 °C was significantly higher than that of the NFT and 4 °C groups (*p* < 0.05), and the MDA content of the 4 °C group was significantly higher than that of the NFT group (*p* < 0.05). This indicates that NFT storage can effectively inhibit the cell membrane damage of fresh *C. pilosula* roots during storage, thereby extending their shelf life.

#### 3.2.3. The Activity of SOD, CAT, PPO and POD

SOD can convert O_2_^−^∙ into H_2_O_2_ and O_2_ via a disproportionation reaction, eliminating the oxidative O_2_^−^∙ damage to cells [29]. From day 0 to day 20 of storage, the SOD activity slowly increased, while from day 20 to day 40, the SOD activity rapidly increased. Compared with the 9 °C, 4 °C and −6 °C groups, the NFT group had the highest SOD activity during storage (Figure 3A).

CAT mainly acts on the high concentrations of H_2_O_2_ in cells and converts them into water and O_2_. As shown in Figure 3B, the CAT activity of the NFT and 4 °C groups slowly increased at the early stage of storage, but rapidly decreased on the 20th day of storage. However, the CAT activity in the NFT and 4 °C groups was significantly higher than that in the 9 °C and −6 °C groups (*p* < 0.05).

Browning in plants is a process in which phenols are oxidized by PPO and POD to form quinones that generate brown pigments. The PPO activity showed an increasing trend, and the PPO activity in the 9 °C and −6 °C groups was significantly higher than that in the NFT and 4 °C groups (*p* < 0.05) (Figure 3C). The POD activity slowly increased during the initial stage of storage and rapidly increased after the 20th day of storage. The PPO activity of the 9 °C and −6 °C groups was significantly higher than that of the NFT and 4 °C groups (*p* < 0.05) (Figure 3D).

#### 3.2.4. The Activity of PG, PME, Cx and β-Glu

The PG, PME, Cx and β-Glu activity rapidly increased at the early stage of storage and decreased after 20 d of storage. However, the increase in these enzymes’ activity was effectively inhibited by NFT and 4 °C storage (Figure 4). Compared with 4 °C storage, NFT storage inhibited the activity of these enzymes at lower levels. On the 40th day of storage, the NFT-stored PG, PME, Cx and β-Glu activities were 82%, 64%, 70% and 53% of those with 4 °C storage, respectively.

#### 3.2.5. Changes in Lobetyolin Content, Syringin Content and Atractylenolide III Content

The content of lobetyolin of the roots showed a slowly decreasing trend, and with the extension in storage time, the advantage of NFT storage in maintaining the lobetyolin content gradually emerged (Figure 5A). The content of syringin also showed a decreasing trend. The content of syringin of the roots in the NFT group remained at a high level, while in the −6 °C group, it remained at a low level (Figure 5B). Except for the slow increase in atractylenolide III content in the 4 °C group on the 40th day of storage, the NFT group, −6 °C group, and 9 °C group all showed a decreasing trend. Among them, the NFT group had the best maintenance effect on the atractylenolide III content (Figure 5C).

#### 3.2.6. Changes in Total Flavonoid Content, Total Phenol Content, and DPPH Radical Scavenging Rate

With the extension in storage time, the content of reactive oxygen species generated by physiological activity increased. For example, respiratory metabolism in the root tissues gradually increased, while the content of total flavonoids and total phenols with antioxidant activity gradually decreased. The DPPH radical scavenging rate gradually decreased, and NFT storage can effectively maintain the total flavonoid, total phenol content and in vitro antioxidant activity of *C. pilosula* roots. The total flavonoid content and DPPH radical scavenging rate of roots at 9 °C showed a significantly faster decreasing trend from the 20th to the 40th day of storage compared to the NFT, 4 °C and −6 °C groups (Figure 5D,F). The total phenol content in the NFT and 4 °C root tissues remained significantly higher than −6 °C and 9 °C (*p* < 0.05) during storage, as shown in Figure 5E.

### 3.3. Multivariate Statistical Analysis

#### 3.3.1. Effects of Near-Freezing Storage on Postharvest Quality of *C. pilosula* Roots Based on PCA

At different storage stages, quantitative data in 20 parameters, including sensory quality, active ingredients, antioxidant capacity and physiological changes, were normalized. Then, the processed data were used for PCA analysis. As shown in Figure 6A, during storage at −6 °C, NFT, 4 °C and 9 °C, the contribution rates of the first and second principal components in the PCA plot of the storage quality and enzyme activity of fresh *C. pilosula* roots were 95.5% and 3.69%, respectively. This indicated that these two principal components basically reflect all the characteristics of all indicators stored at different temperatures. The aggregation of −6 °C, NFT, 4 °C and 9 °C on the 20th day of storage indicated that the first and second principal components of these four samples are similar. The −6 °C, NFT, 4 °C and 9 °C groups on the 20th day of storage were similar to the NFT, and 4 °C groups on the 40th day of storage, indicating that the storage quality and enzyme activity of these six samples were similar. In addition, the distance between the −6 °C, NFT, 4 °C and 9 °C groups on the 20th day of storage, as well as the NFT group on the 40th day of storage, and fresh samples was relatively close, indicating that the quality of roots on the 20th day of storage was similar to that of fresh samples. NFT storage can effectively maintain the storage quality of fresh *C. pilosula* roots and extend their shelf life.

#### 3.3.2. Effects of Near-Freezing Storage on Postharvest Quality of *C. pilosula* Roots Based on OPLS-DA

From the VIP plot (Figure 6B), it can be seen that the difference indicators with a VIP value greater than 1 are the content of atractylenolide III, POD activity, DPPH radical scavenging rate, lobetyolin content, total flavonoid content, syringin content, weight loss, total phenol content and sensory score. These indicators were the main landmark indicators for evaluating the quality differences in *C. pilosula* roots stored at −6 °C, NFT, 4 °C and 9 °C at different stages and fresh samples. The further away from the center point in the S-plot (Figure 6C) the indicator was, the more prominent its differentiation contribution to the storage effect of fresh samples and the roots stored at −6 °C, NFT, 4 °C and 9 °C was, with a higher corresponding VIP value. Among them, the indicators in the lower-left corner had a higher level in the fresh samples, such as the content of atractylenolide III, DPPH radical scavenging rate, lobetyolin content, total flavonoid content, syringin content, etc. This also suggested the rationality of using fresh *C. pilosula* roots directly as food and pharmaceutical raw materials.

The indicators in the upper right had higher levels in roots stored at different temperatures, such as POD activity and weight loss. This indicated that POD activity and weight loss can be used as the main indicators to evaluate the quality deterioration of *C. pilosula* roots during storage.

#### 3.3.3. Effects of Near-Freezing Storage on Postharvest Quality of *C. pilosula* Roots Based on Cluster and Correlation Analysis

In order to determine the relationship between the quality of *C. pilosula* roots and enzyme activity indicators stored at different temperatures, 20 characteristic indicators were analyzed using cluster heatmaps (Figure 6D). The results showed that the fresh roots were clustered into a single group, that is, different temperature storage had varying degrees of effects on the *C. pilosula* roots. The samples stored at −6 °C, 4 °C, and 9 °C for 20 and 40 d were clustered into two categories, indicating that the storage quality and enzyme activity of the roots changed significantly over time. The samples stored in NFT for 20 and 40 d are grouped together, indicating that NFT storage is beneficial for maintaining the stable and controllable quality of *C. pilosula* roots. There are 20 characteristic indicators that were mainly divided into two categories. Cluster 1 was composed of respiratory intensity and nine indicators that were positively correlated with the storage quality of the *C. pilosula* roots, including total phenol content, syringin content, sensory score, CAT activity, total flavonoid content, lobetyolin content, DPPH radical scavenging rate, SOD activity and atractylenolide III content. They had high levels in the fresh samples and significantly decreased after 40 d of storage at −6 °C, 4 °C and 9 °C. Cluster 2 included weight loss and nine key enzymes involved in physiological deterioration processes such as oxidative damage, cell wall degradation and browning, which are enhanced after 20 d of storage at −6 °C, 4 °C and 9 °C.

Pearson’s correlation analysis was used to evaluate the interaction between different indicators. As shown in Figure 6E, the sensory score was positively correlated with the respiratory intensity, SOD activity, CAT activity, Cx activity, lobetyolin content, syringin content, atractylenolide III content, total flavonoid content, total phenol content and DPPH radical scavenging rate, indicating that NFT storage may mainly maintain high sensory quality by increasing antioxidant enzyme activity and non-enzymatic antioxidant substance contents in *C. pilosula* roots. The weight loss and O_2_^−^∙ production rate, H_2_O_2_ content, MDA content and PPO, POD, PG, PME, Cx and β-Glu activity were positively correlated, indicating that NFT storage may mainly reduce weight loss by inhibiting oxidative damage and the cell wall degradation of *C. pilosula* roots.

## 4. Discussion

Usually, a faster respiratory metabolism accelerates the consumption of nutrients and the generation of ROS. When cells lose their homeostasis, the excessive accumulation of ROS leads to oxidative stress, which further leads to oxidative damage to the cell membrane [30]. However, changes in the membrane lipid composition during low-temperature frostbite and aging are similar. In this experiment, the 9 °C group of *C. pilosula* roots maintained a high level of respiratory intensity, O_2_^−^∙ production rate and H_2_O_2_ content throughout the entire storage process. At the same time, the MDA content rapidly increased, significantly higher than the −6 °C, NFT and 4 °C groups at the end of storage, indicating that the roots underwent rapid oxidative aging during storage at 9 °C. The respiratory intensity of roots in the −6 °C group remained low during storage, but the MDA content remained high, indicating that root frostbite also accelerated cell membrane damage. We found that NFT storage can reduce ROS accumulation by inhibiting the respiratory intensity, thereby alleviating cellular oxidative damage during the storage of roots. The clearance of ROS in the cells mainly relies on the action of non-enzymatic antioxidants and antioxidant enzymes. For example, 1-MCP treatment can inhibit the respiratory metabolism of edible roses while increasing the content of total phenols and anthocyanins and the activity of SOD and CAT to maintain storage quality [31]. There are similar conclusions in the study of ginger storage [32]. Lobetyolin, syringin, atractylenolide III, flavonoids and phenols are all substances related to the stress resistance and antioxidant activity of *C. pilosula* roots [33,34,35]. The content of them was the highest in the fresh samples. During 40 d of storage, the content of the five active components showed varying degrees of decline, while NFT storage had a significant advantage in maintaining their content. This indicated that NFT storage may enhance the antioxidant capacity of *C. pilosula* roots by increasing their antioxidant enzyme activity and active component content, thereby extending their shelf life to the 40th day.

The problem of the color of *C. pilosula* roots during postharvest storage gradually deepening seriously affects consumers’ sensory acceptance. The appearance change in fresh agricultural products is mainly caused by an enzymatic browning reaction, which refers to the process of phenols being oxidized by PPO and POD to form quinones, thereby generating brown or black pigments [36]. During this process, oxygen, phenols and oxidase are key factors, and the excessive accumulation of ROS leads to oxidative damage to the cell membrane, causing them to interact and react with each other [37]. Inhibiting the activity of oxidase is the most direct method to delay enzymatic browning, such as in Li X [38], which combined modified atmosphere packaging and low-temperature refrigeration to delay the browning of freshly cut lilies by inhibiting the PPO and POD enzyme activity. In addition, improving the antioxidant capacity of fresh crops is also of great significance in inhibiting enzymatic browning. Ali et al. [39] found that oxalic acid treatment with 10 mmol·L^−1^ significantly increased the total phenol and ascorbic acid content, as well as the SOD and CAT activity of freshly cut lotus roots. In this experiment, NFT storage not only increased the content of antioxidant substances and antioxidant enzyme activity in roots but also showed inhibitory effects on the PPO and POD activity. This is consistent with the result that NFT storage can better maintain the bright color of *C. pilosula* roots.

Softening is an irreversible part of the maturation process of *C. pilosula* roots, and fresh roots have a fresh and juicy texture. However, excessive softening reduces the value of the product while increasing the possibility of microbial infection, which is an important quality deterioration problem that needs to be addressed during postharvest storage. Softening is a result of the plant cell wall metabolism. According to existing research reports [40], the cell wall of *C. pilosula* roots is composed of cellulose and pectin polysaccharides. Studies have shown that PME and PG can cause the demethylation of pectin polysaccharides, which is an important reason for fruit softening during ripening and postharvest storage [41]. β-Glu is a type of cellulase that has the ability to hydrolyze the plant cell wall skeleton cellulose. Numerous studies such as storage research on cantaloupe [42], lychee [43], peach [44] and mango [45] have shown that reducing the activity of the cell-wall-degrading enzymes mentioned above can effectively inhibit the cell wall metabolism in vegetables and fruits, thereby delaying the occurrence of excessive softening. In this experiment, the PG, PME, Cx and β-Glu activity showed a trend of first increasing and then decreasing, and the activity of the four degrading enzymes in the *C. pilosula* roots during NFT storage remained at a relatively low level, consistent with the result that NFT storage could maintain its hard and elastic texture during storage.

## 5. Conclusions

In this experiment, postharvest *C. pilosula* roots were stored in NFT (−1 °C) and traditional low temperatures (−6 °C, 4 °C, and 9 °C) for 40 d, respectively. Then, 20 quality parameters related to sensory quality, active component content, antioxidant capacity and physiological changes in the root samples were quantitatively measured every 20 d to monitor the overall quality changes. The PCA and OPLS-DA results of quantitative data could well describe the comprehensive quality characteristics of different root samples and clearly distinguish them. As the storage time prolongs, the overall quality differences in the roots increase. However, the overall quality of the roots stored at NFT and 4 °C was closer to the fresh samples, especially for NFT storage. Correlation analysis further indicated that NFT storage could effectively improve the antioxidant enzyme activity and active component content, as well as reducing the activity of browning-related enzymes and cell-wall-degrading enzymes to maintain the sensory quality of roots. In summary, both NFT and 4 °C storage can delay the quality deterioration process of roots, such as oxidative aging, browning and softening, to varying degrees. In particular, NFT storage can effectively maintain a high level of active component content and antioxidant capacity until the 40th day of storage, making it a more suitable storage temperature for the processing of *C. pilosula* roots in the production area after harvest. This study has combined chemometrics with the quantitative determination of quality parameters to construct a new integrated quality evaluation method from four aspects: sensory quality, active components, antioxidant capacity and physiological changes. This is used to evaluate and distinguish the overall quality of *C. pilosula* roots during postharvest storage. This provides a new approach to the quality evaluation of medicinal herbs after harvest. At the same time, the potential of improving the quality of fresh medicinal materials by regulating storage conditions such as temperature was emphasized. More importantly, this study provides a theoretical and data-driven basis for the selection of storage conditions and the development of preservation techniques during the processing, storage and transportation of fresh *C. pilosula* roots in their production areas. It also provides reference for the preservation and storage research of other Chinese medicinal materials.

## Figures and Tables

**Figure 1 foods-12-04418-f001:**
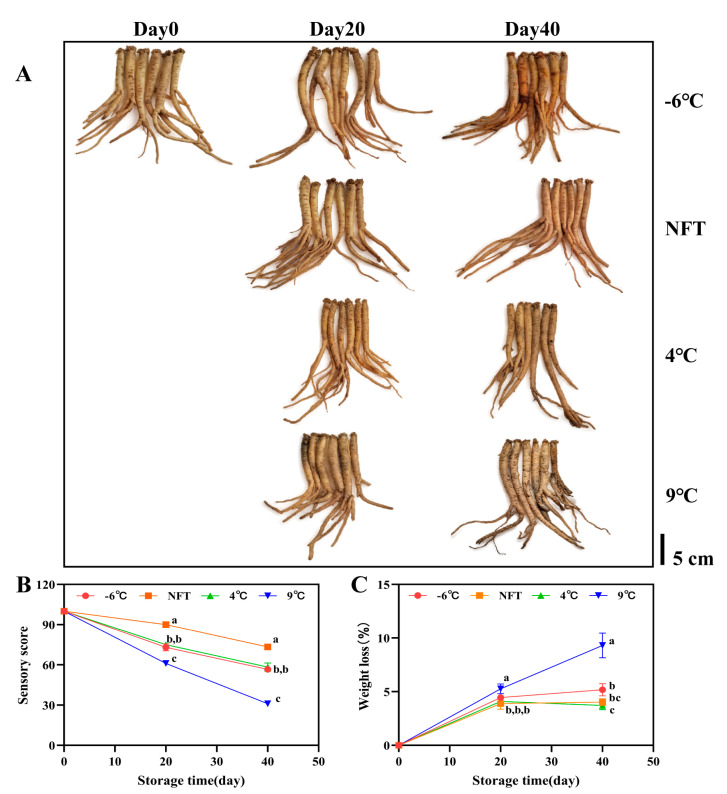
Changes in appearance (**A**), sensory score (**B**) and weight loss (**C**) of fresh *C. pilosula* roots during storage at −6 °C, near -freezing temperature [NFT (−1 °C)], 4 °C and 9 °C. The error bar represents the standard deviation of three replicates and different letters at the same time point indicate significant differences (*p* < 0.05).

**Figure 2 foods-12-04418-f002:**
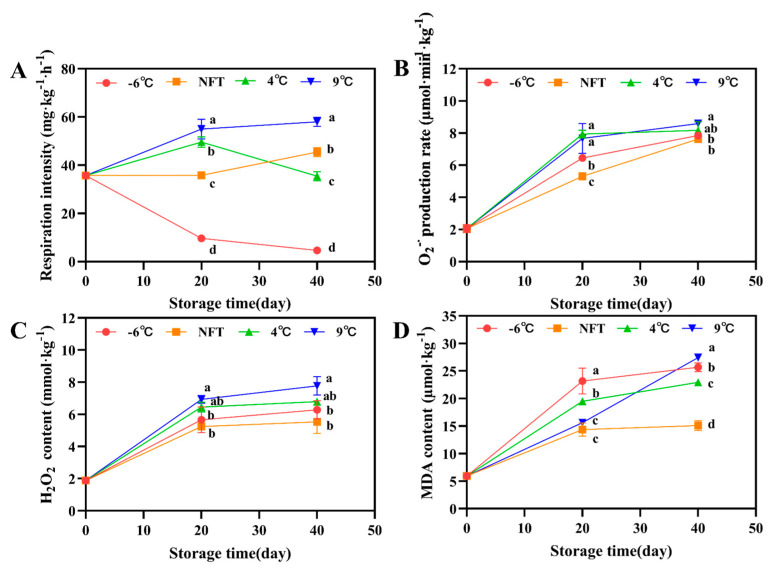
Changes in respiratory intensity (**A**), O_2_^−^∙ production rate (**B**), H_2_O_2_ content (**C**) and MDA content (**D**) of fresh *C. pilosula* roots during storage at −6 °C, NFT, 4 °C and 9 °C. The error bar represents the standard deviation of three replicates and different letters at the same time point indicate significant differences (*p* < 0.05).

**Figure 3 foods-12-04418-f003:**
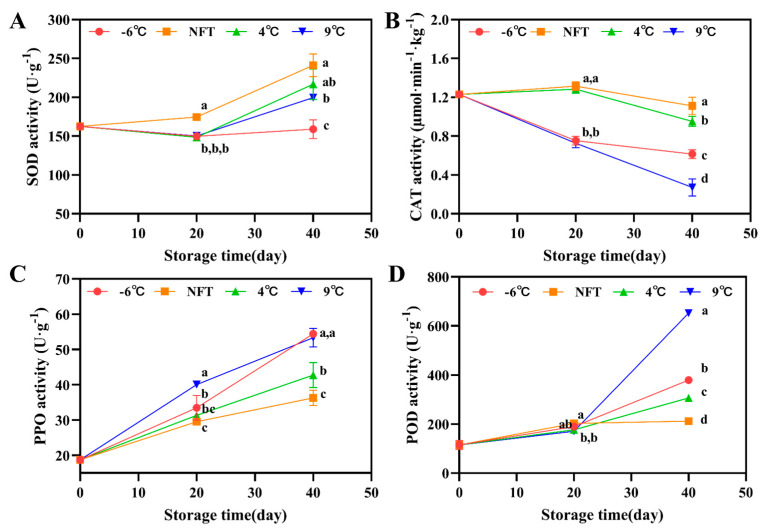
Changes in SOD activity (**A**), CAT activity (**B**), PPO activity (**C**) and POD activity (**D**) of fresh *C. pilosula* roots during storage at −6 °C, NFT, 4 °C and 9 °C. The error bar represents the standard deviation of three replicates and different letters at the same time point indicate significant differences (*p* < 0.05).

**Figure 4 foods-12-04418-f004:**
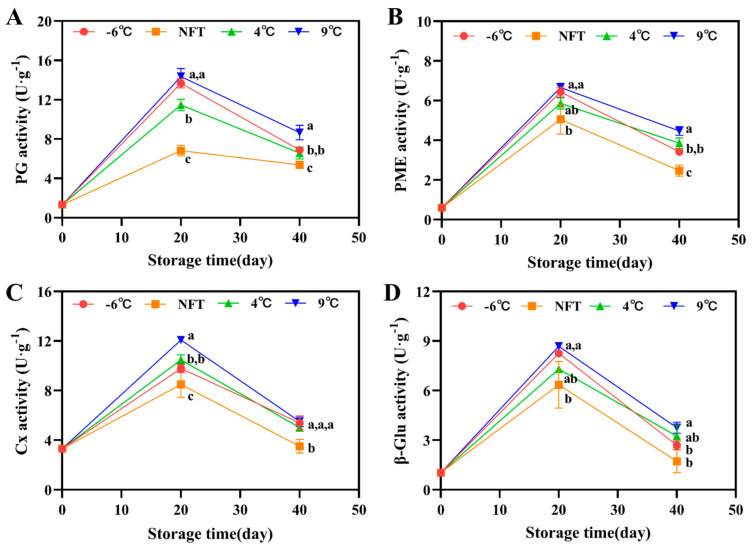
Changes in PG activity (**A**), PME activity (**B**), Cx activity (**C**) and β-Glu activity (**D**) of fresh *C. pilosula* roots during storage at −6 °C, NFT, 4 °C and 9 °C. The error bar represents the standard deviation of three replicates and different letters at the same time point indicate significant differences (*p* < 0.05).

**Figure 5 foods-12-04418-f005:**
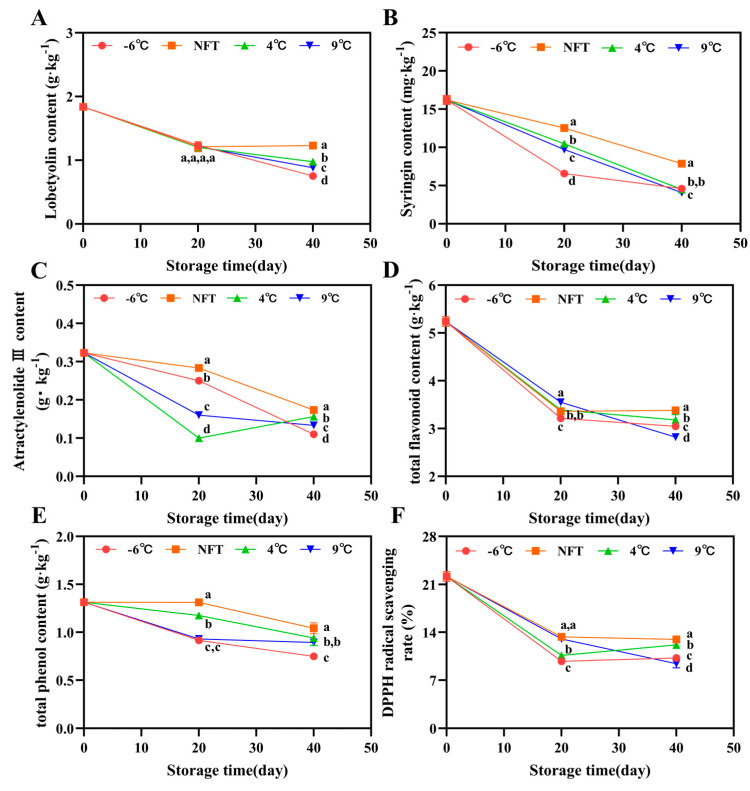
Changes in lobetyolin content (**A**), syringin content (**B**),atractylenolideIII content (**C**), total flavonoid content (**D**), total phenol content (**E**) and DPPH radical scavenging rate (**F**) of fresh *C. pilosula* roots during storage at −6 °C, NFT, 4 °C and 9 °C. The error bar represents the standard deviation of three replicates and different letters at the same time point indicate significant differences (*p* < 0.05).

**Figure 6 foods-12-04418-f006:**
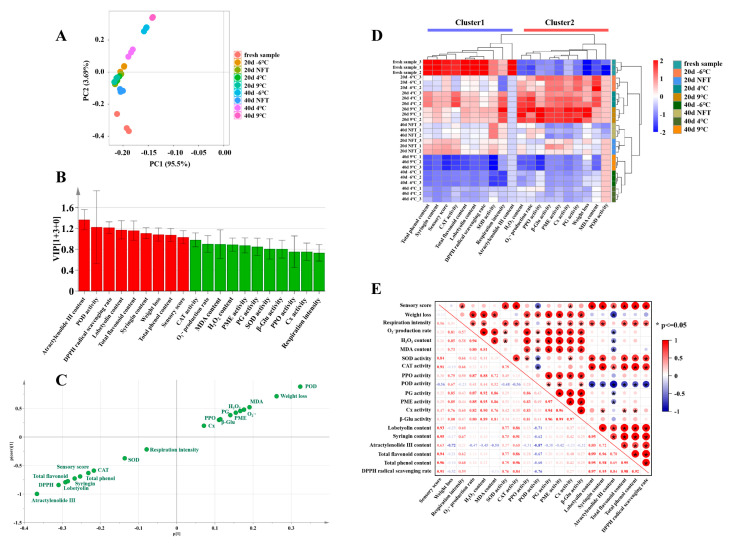
PCA analysis (**A**), VIP plot (**B**), S-plot (**C**), cluster heatmap (**D**) and Pearson’s correlation analysis (**E**) of storage quality and enzyme activity of fresh *C. pilosula* roots during storage at −6 °C, NFT, 4 °C and 9 °C. Red indicates a VIP value greater than 1, while green indicates a VIP value less than 1.

## Data Availability

Data are contained within the article.

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
