# Peer review of "Exploration of Postharvest Conditions for Codonopsis pilosula Nannf. var. modesta (Nannf.) L. T. Shen Roots Based on Sensory Quality, Active Components, Antioxidant Capacity and Physiological Changes at Different Storage Temperatures"

_foods, 2023, doi:10.3390/foods12244418_

Round 1

Reviewer 1 Report

Comments and Suggestions for Authors

The topic is interesting, and the conclusions can help the industry to optimise its processes.

However, I have some concerns about the methodology that compromise the validity of the results:

-        - The authors state that the sensory evaluation was carried out by a panel of experts. I have serious doubts about this.

Were the people on the panel specifically trained to evaluate Codonopsis pilosula roots? How many hours? What were the standards used for training? Why use 30 people when a panel of experts usually consists of no more than 8 people?

If they haven't received specific training, I suggest you comment on this. Perhaps you can refer to them as a "semi-trained panel".

If they were really an expert panel, you should comment on the number of hours of training they had.

On the other hand,  the way you have processed the sensory data is not appropriate. Although it was not explained in the materials and methods section (you should do it), I noticed that you obtained the summatory of the different attributes (colour, taste, etc.). This is a wrong approach. Just imagine that one of the treatments resulted in roots with a very bad texture, but very good scores in the other parameters. You will conclude that the quality is good, which would not be true. I mean, you need to evaluate all the parameters independently, as one of them may be the limiting one.

-        There is a lack of information about the statistical analysis. Based on the results of the figures, I am afraid that the analytical data were not pre-processed before the PCA was performed. This is a necessary step because the data come from different parameters and need to be normalised before the PCA. For example, the range of CAT activity data is from 0 to 1.2, while the range of POD is from 100 -700. This is the reason why you need to normalize the data before PCA.

On the other hand, the way the article is written is also a problem. The methodology is written in the present tense, as if giving a cooking recipe. You should write it in past tense. The format of the citations in the text need to be revised.

Comments on the Quality of English Language

The methodology is written in the present tense, as if giving a cooking recipe.

Reviewer 2 Report

Comments and Suggestions for Authors

I have reviewed the manuscript entitled: “Exploration of storage conditions for postharvest Codonopsis pilosula based on integrated quality evaluation of sensory quality, active components, antioxidant capacity and physiological changes”. The manuscript evaluates the effect of temperature conditions during storage on sensory quality, effective components contents, antioxidant capacity, and postharvest physiological quality changes of C. pilosula roots and the obtained date are evaluated in terms of quality characteristics of roots.

The abstract is well written and summarizes the main findings of the study.

Introduction is well addressed and clear.

Materials and methods section need improvement. The methods should be better presented, in a more scientific way, rather than the way that the authors chose to present (looks like bullet points activities).  

The discussion is clear, and I believe it can be improved if more studies are added, in order to support the findings of this study.

The conclusions are just the presentation of the main results. They should also emphasize the main ideas of the manuscript and the relevance of the work.

Comments on the Quality of English Language

English language requires minor revision, for a better understanding of the study.

Reviewer 3 Report

Comments and Suggestions for Authors

In the present manuscript, the Authors describe a postharvest storage experiment set up to define the most appropriate conditions for the maintenance of Codonopsis pilosula roots quality. This species is very appreciated in traditional Chinese medicine for its properties. In particular, the Authors compared near-freezing temperature (NFT) to the conventionally used temperatures in a 40-day storage.

The work is conducted with care and the paper is overall well written. Please, check the language errors, especially regarding the use of verbs. In my opinion, the paper needs some revision.

Specific comments are as follows:

 Introduction

Please, explain “qi”, not everybody is familiar with traditional medicine

Line 82: please change “stoichiometric” to “multivariate”

Materials and methods: please describe the panel composition (gender, age).

Line 131: Please insert “analyzed” after “statistically”

2.3: which instrument was used for spectrophotometric measurements?

2.3.6: This analysis is very important, so please add some more description to the references citation

Lines 216-217: Please specify, the % of what? Even if you refer to a published method, the system should be calibrated, If possible, using standard solutions at known concentrations, and the DPPH scavenging activity should be expressed as standard equivalents/weight of the sample, in order to facilitate the comparison with literature data. 

Comments on the Quality of English Language

There are some errors regarding the use of verbs. Please, check.

Round 2

Reviewer 1 Report

Comments and Suggestions for Authors

Thank you for the revised version.I see that you have tried to improve the manuscript. 

This is the weak point of your study is the sensory analysis. I suggest you contact an expert in sensory science for future studies. Nevertheless, I think you have arrived at relevant results as a whole.

Some changes are needed before publication: 

Add the NFT temperature in the abstract: -1.2ºC

Please remove Table S1. Many of the attributes are not in the appropriate category. I can´t accept the manuscript with this table. 

Flavor is not odor

Crispy is not taste, it is texture

You use 'hardness' instead of 'texture

Brittle is texture, not taste

I think it is better to remove the table and just comment in material and methodds section that you used a 1-25 scale where 25 is high quality and 1 is poor quality for:  appearance, odor, texture and taste. 

Reviewer 3 Report

Comments and Suggestions for Authors

The Authors carefully answered all my previous remarks. In my opinion, the manuscript is suitable for publication in Foods.

Author Response

Dear reviewer,

We feel great thanks for your review work. Thank you again for all your helpful comments and suggestions. They have inspired many new analyses that we believe have led to a stronger paper.